# Dysmetabolism and Neurodegeneration: Trick or Treat?

**DOI:** 10.3390/nu14071425

**Published:** 2022-03-29

**Authors:** Adriana M. Capucho, Ana Chegão, Fátima O. Martins, Hugo Vicente Miranda, Sílvia V. Conde

**Affiliations:** CEDOC, NOVA Medical School, Faculdade de Ciências Médicas, Universidade NOVA de Lisboa, 1150-082 Lisboa, Portugal; adriana.capucho@nms.unl.pt (A.M.C.); ana.chegao@nms.unl.pt (A.C.); fatima.martins@nms.unl.pt (F.O.M.); hmvmiranda@nms.unl.pt (H.V.M.)

**Keywords:** insulin signaling, metabolic disorders, hypercaloric diets, neurodegeneration

## Abstract

Accumulating evidence suggests the existence of a strong link between metabolic syndrome and neurodegeneration. Indeed, epidemiologic studies have described solid associations between metabolic syndrome and neurodegeneration, whereas animal models contributed for the clarification of the mechanistic underlying the complex relationships between these conditions, having the development of an insulin resistance state a pivotal role in this relationship. Herein, we review in a concise manner the association between metabolic syndrome and neurodegeneration. We start by providing concepts regarding the role of insulin and insulin signaling pathways as well as the pathophysiological mechanisms that are in the genesis of metabolic diseases. Then, we focus on the role of insulin in the brain, with special attention to its function in the regulation of brain glucose metabolism, feeding, and cognition. Moreover, we extensively report on the association between neurodegeneration and metabolic diseases, with a particular emphasis on the evidence observed in animal models of dysmetabolism induced by hypercaloric diets. We also debate on strategies to prevent and/or delay neurodegeneration through the normalization of whole-body glucose homeostasis, particularly via the modulation of the carotid bodies, organs known to be key in connecting the periphery with the brain.

## 1. Introduction

Aging is broadly defined as the time-dependent functional decline that affects most living organisms. It is characterized by a progressive loss of physiological integrity, leading to impaired function and increased vulnerability to death [1]. Aging is pointed as the main risk factor for major human pathologies, including cancer, diabetes mellitus (DM), cardiovascular disorders, and neurodegenerative diseases. The hallmarks of aging are genomic instability, telomere attrition, epigenetic alterations, loss of proteostasis, deregulated nutrient sensing, mitochondrial dysfunction, cellular senescence, stem cell exhaustion, and altered intercellular communication [1]. Aging is therefore accompanied by a variety of physiological changes that make the organisms highly prone to so-called aged-related disorders. On the other hand, several diseases such as cardiometabolic and metabolic disorders can accelerate the aging and senescence process.

The metabolic syndrome is defined as a cluster of disorders including obesity, DM, and cardiovascular disease, and is reaching pandemic levels worldwide. Several studies have shown that the prevalence of metabolic diseases increases up to 50% in people aged over 65 years [2,3]. Consequently, as the global population ages, there will be a higher prevalence of obesity, DM, and cardiovascular disease [4,5].

Additionally, neurodegenerative diseases, which are characterized by the progressive loss of the structure or function of neurons, are classified as an age-related cluster of pathologies that were shown to be increased in the set of metabolic disorders [6,7]. Age and metabolic disorder-related neurodegenerative diseases, such as Alzheimer’s (AD) and Parkinson’s (PD), are increasing worldwide, and it is not clear whether metabolic syndrome is the cause or the consequence of these neurodegenerative diseases [7,8,9,10].

Changing lifestyles and eating habits opened a door to pronounced alterations in the aging process. In this review, we mainly focus on the relationship between metabolic syndrome and neurodegenerative diseases, exploring the role of hypercaloric diets in brain function and the physiological role of insulin and insulin resistance in the central nervous system (CNS).

## 2. Insulin and Insulin Signaling Pathways

Around one hundred years ago, in 1921, an important hormone—insulin—was discovered by Frederick G. Banting, Charles Best, and John MacLeod in Canada. This event completely changed the lives of mankind, and since then thousands of lives have been saved [11].

Insulin is a 51 amino acids peptide hormone, organized in 2 polypeptides chains, A (21 amino acids) and B (30 amino acids), which are connected by disulfide bonds. Insulin production and release is carried out by pancreatic β-cells. This hormone plays a fundamental role in regulating fat and carbohydrate metabolism, promoting their conversion into storage macromolecules such as glycogen, proteins, and lipids [12,13,14].

Insulin is the most important anabolic hormone in the organism and plays a critical role in the regulation of different physiological processes, including metabolism, cell growth and differentiation [12,15]. The effects of insulin are highly pleiotropic and differ according to the target tissue. Concerning the peripheral regulation of metabolism, the major insulin-sensitive tissues include the skeletal muscle, adipose tissue, and liver [16]. Insulin stimulates the uptake of glucose in the skeletal muscle and adipose tissue, inhibits hepatic glucose production, and promotes the storage of substrates in the adipose tissue, liver, and muscle by triggering lipogenesis, glycogen, and protein synthesis, and by inhibiting lipolysis, glycogenolysis, and protein breakdown [15,16,17]. Insulin also plays a role in the CNS, mainly controlling appetite, by acting in the hypothalamus, one of the most important neuronal centers to the control of satiety and feeding through negative feedback to ensure balanced energy homeostasis [18].

Although insulin plays several functions in the organism, it is widely known as a glucose homeostasis regulating hormone. The tight control of the plasma levels of glucose is driven by the balance between glucose absorption from the intestine after feeding, production by the liver and uptake and utilization by peripheral tissues. An increase in the plasma glucose levels after a meal triggers insulin production in the pancreatic islets by β-cells.

Insulin acts by binding to its receptor, the insulin receptor (IR). IR is a glycoprotein that belongs to the tyrosine kinase receptor family, being composed by an α extracellular and β transmembrane subunits. When insulin binds to the α subunit, it triggers the dimerization of the receptor, forming an α2β2 complex, leading to the autophosphorylation of β subunit at Tyr1158, 1162, and 1163, the so-called activation of IR phosphorylation cascade (Figure 1) [15].

Activation of IR signaling pathway leads to the recruitment and phosphorylation of several proteins such as the insulin receptor substrates (IRS 1-4), that will induce the activation of intracellular pathways as phosphatidylinositide-3-kinase (PI3K) and the mitogen-activated protein kinase (MAPK) cascades [15,19]. Insulin-induced activation of Ras-MAPK promotes the regulation of cell growth and mitogenesis, whereas PI3K activation generates phosphatidylinositol (3,4,5)-triphosphate (PIP3) that will activate phosphoinositide dependent protein kinase-1 and -2 (PDK1 and PDK2), which regulate the effect of insulin on metabolism and pro-survival. Also, PDK1 and 2, have an important function in the activation of Protein Kinase B (AKT/ PKB) [15,17,20]. AKT has a key function in phosphorylating several downstream proteins, as glycogen synthase kinase b (Gsk3b), that inhibits glycogen synthesis [20,21]. AKT also phosphorylates other mediators such as the activation of AKT substrate 160 kDa (AS160) and activates Rab10 GTPase that leads to the translocation of the glucose transporter 4 (GLUT4) to the plasma membrane to promote glucose uptake [22]. Phosphodiesterase 3B (PDE3B), an enzyme that catalyzes the degradation of cyclic adenosine monophosphate (cAMP), is also activated by AKT. Additionally, AKT inhibits cAMP response element-binding protein (CREB)-regulated transcription coactivator 2 (CRTC2), important to increase hepatic gluconeogenesis. AKT also triggers liver lipogenesis by phosphorylating sterol regulatory element-binding protein 1 (SREBP1) [23], and phosphorylates Forkhead box protein O1 (Foxo1), inhibiting its transcriptional activity and leading to a suppressed liver glucose production [15,21].

## 3. Metabolic Syndrome: Insulin Resistance and Diabetes

Failure in insulin production, insulin tissue-sensitivity, and/or insulin signaling is a pathophysiological hallmark of pre-diabetic states, resulting in hyperglycemia due to impaired uptake of glucose by cells and deregulation of hepatic glucose production [24]. Furthermore, insulin resistance together with obesity, hypertension, hyperlipidemia, dyslipidemia and hyperglycemia are key features of metabolic syndrome. DM is one of the most prevalent metabolic disorders, characterized by chronic hyperglycemia.

DM has reached pandemic levels, affecting more than 463 million people and being among the top 10 causes of deaths in adults worldwide [25]. DM is a complex, chronic condition that has a major impact on the lives and well-being of individuals, families, and a vast associated economic burden for societies [26]. DM refers to a group of chronic metabolic diseases characterized by hyperglycemia and dysfunction or destruction of pancreatic β-cells, causing defects in insulin secretion, insulin action, or both [27,28,29]. The heterogeneous etiopathology mechanisms leading to the decline in function, or the complete destruction of β-cells includes genetic predisposition and abnormalities, epigenetic processes, insulin resistance, autoimmunity, concurrent illnesses, inflammation, and environmental factors [27,30]. Moreover, DM is also associated with disturbances of carbohydrate, fat, and protein metabolism.

The World Health Organization (WHO) has recently proposed the classification of DM into six subtypes: type 1 diabetes mellitus (T1D); type 2 diabetes mellitus (T2D); hybrid forms of DM; other specific types; unclassified DM; hyperglycemia during pregnancy [31]. T1D is characterized by the destruction of pancreatic β-cells, resulting in the inability of the body to produce insulin. Consequently, individuals suffering from T1D are insulin dependent. In contrast, T2D which accounts for 90% of all diabetic cases, is characterized by insulin resistance. The pathogenesis of T2D involves abnormalities in both peripheral insulin action and insulin secretion by pancreatic β-cells, resulting in hyperglycemia. Insulin resistance, in an initial phase called prediabetes, is usually compensated by hyperinsulinemia, that, although it might be tolerated in the short term, will lead to a vicious cycle in which chronic hyperinsulinemia exacerbates insulin resistance and contributes directly to β-cells exhaustion and the definitive settlement of T2D.

DM predisposes to long-term complications including retinopathy, nephropathy, and neuropathy. DM increases the risk for cardiovascular diseases, obesity, cataracts, erectile dysfunction, and non-alcoholic fatty liver disease (NAFLD) [31].

Furthermore, insulin resistance syndrome negatively impacts the CNS function and DM is associated with an increased risk to develop neuropsychiatric conditions and neurodegenerative disorders [6]. As such, the control of peripheral and central insulin levels can be an important approach to prevent the development of those pathologies.

### Impact of Hypercaloric Diets on Insulin Resistance and Metabolic Syndrome

The consumption of hypercaloric diets and overnutrition is a common trigger for the development of obesity, which is the major driver to insulin resistance, prediabetes and T2D. In recent years, the role of nutrition and lifestyle in the development of metabolic diseases has been extensively explored. Due to the fast-paced and sedentary lifestyles of modern society, the intake of hypercaloric and unhealthy food has risen, increasing the incidence of metabolic diseases in the population, and negatively impacting the aging process, aggravating the aging-associated consequences [25,31].

It is well known that the amount and nature of macro and micronutrient consumption is associated with the development of obesity and T2D, since the dietary composition and volume could be implicated in insulin resistance states, being in the genesis of T2D [32]. More than the hypercaloric composition, overnutrition is also one of the causes to the development of dysmetabolism [33]. Excessive or incorrect nutrients consumption, both carbohydrates or fat, promotes a general state of inflammation, which has an important role in insulin dysfunction as well as in energy production and catabolism of those nutrients [34].

Studies, both in animal models and humans, showed that hypercaloric diets, independently of their composition, lead to dysmetabolic states. For example, diets enriched in fat, free sugars, and fructose lead to whole-body insulin resistance [32,35]. However, different diets composition originates different metabolic features [36]. Melo et al. [36], by submitting different groups of Wistar rats to a high-fat diet (HF), high-fat–high-sucrose diet (HFHSu) and high-sucrose diet (HSu) for different periods of time, showed that the animals submitted to a HF diet for 19 weeks exhibited increased weight gain when compared to the other groups. On the other hand, animals fed with an HSu diet for 16 weeks did not exhibit a significant increase in weight gain, while animals under HFHSu diet showed an intermediate increase in weight gain between the HSu and HF groups. In fact, Melo et al. showed that the HF diet feeding model was the hypercaloric diet model that presented more phenotypic features related to T2D and obesity in humans, since they developed insulin resistance, impaired lipid and glucose metabolism, elevated sympathetic activity, hypertension, hypertriglyceridemia, and increased lipid deposition in the liver resembling NAFLD [37]. In the case of the HFHSu diet, animals exhibit significant alterations in insulin and C-peptide levels, which can be associated with impaired pancreatic insulin secretion and/or impaired insulin clearance. This finding suggests that the combination of sucrose with fat has a greater impact in glucose metabolism and insulin secretion [36,38]. These differences in the effects of diet nutrient composition over metabolism needs more clarification for a full understanding of the mechanisms in physiological and pathophysiological conditions, as well as for developing new therapies for these types of pathologies.

The adipose tissue is the most important organ for fatty acids storage. However, adipocytes have a limited capacity to accumulate fat, and when they over-expand, they create a hypoxic environment. This environment triggers the overactivation of hypoxic inducible factor 1 (HIF-1) protein creating an inflammatory state that leads to insulin resistance in the adipocytes [39,40,41]. Additionally, insulin has a role as anti-lipolytic hormone, therefore inhibiting the release of adipose tissue fatty acids to the circulation [42]. However, in a state of excessive fat accumulation within the adipose tissue and a state of insulin resistance, the levels of free fatty acids in the circulation increase. This will stimulate the uptake of fatty acids by other tissues, such as the liver, which does not have the capacity to store high amounts of fat. This increased uptake of free fatty acids by the liver contributes to liver insulin resistance and consequent hepatic glucose production and glycogen synthesis deregulation, aggravating the whole-body insulin resistance and glucose homeostasis impairment [43,44]. This event is one of the first consequences of dysmetabolism, leading to the development of NAFLD, which is present in 90% of obese T2D patients [45].

Regarding the role of sugar on metabolism and dysmetabolism, it is known that the different types of sugars are additive, palatable, and rewarding compounds from food [46]. Excessive sugar consumption leads to several metabolic disorders, and some related brain pathologies, impairing reward systems which may lead to compulsive eating [47,48].

Sugar refers to a category of carbohydrates which includes fructose and glucose (monosaccharides), and lactose and sucrose (disaccharides), having different roles in the organism. Many people consider fructose as a healthy sugar because it is naturally present in fruit and other vegetables. However, our body does not respond similarly to the fructose that is present in the fruit and the fructose that is added on the diet [49,50,51]. In fact, the added fructose, mainly in the form of corn syrup, is one of the major causes of insulin resistance and impaired lipid metabolism associated to T2D and its comorbidities [49,50,51,52,53]. Additionally, recent studies have shown that when fructose is replaced by glucose in starch form in the diet these pathological features were alleviated [54,55]. The differences in the metabolic features promoted by glucose and fructose indicate that the metabolic pathways involved are different. In fact, fructose absorption in the gut promoted by GLUT5 is independent of insulin as well as its usage in the liver. In the liver, fructose is converted into glucose, lactate, and fatty acids, that will be released to the blood, oxidized, and used by other tissues [51,56,57]. Glucose is taken by muscle and the adipose tissue to be used as energy source. In contrast, these tissues do not uptake fructose since this sugar is not a primary source of energy [46]. In opposition to glucose and sucrose, fructose promotes smaller effects in plasma glucose levels, satiety hormones and insulin action, leading to overconsumption of calories to promote satiety [58].

## 4. The Role of Insulin in the Brain

As stated before, insulin acts in insulin-sensitive peripheral tissues such as the liver, the skeletal muscle, the pancreas, and the adipose tissue to regulate glucose homeostasis. However, insulin has been also described to play a key role in the CNS [19,20]. Insulin action in the brain regulates several processes including energy expenditure, glucose homeostasis, feeding behavior and satiety, reward pathways, reproduction, cell proliferation and differentiation. Moreover, insulin has neuroprotective and neuromodulatory properties and plays a crucial role in neuronal transmission and survival, neurogenesis, plasticity, and memory and cognition [19,20,59,60].

Insulin reaches the brain from circulation by crossing the blood–brain barrier (BBB) through a selective, saturable transport [19,61,62]. Although insulin levels in cerebrospinal fluid (CSF) are much lower than in plasma, these levels are correlated, indicating that most insulin in the brain may derive from circulating pancreatic insulin [63,64]. Some authors demonstrated the presence of relatively high concentrations of insulin in brain extracts, suggesting that insulin could also be synthetized in the CNS and released locally, as already established for other hormones [65,66]. In fact, some data supports that neurons and astrocytes may produce insulin in relative higher amounts [67,68,69]. Insulin synthesis was described to be independent of its peripheral concentrations, since insulin levels in the hypothalamus were not related to pancreatic insulin concentrations during fasting [70]. Additionally, it was found that GABAergic neurons synthetize insulin in the rat cerebral cortex, as well as in the hypothalamus and hippocampus [71]. However, insulin production by neurons and glial cells is still under debate.

Nevertheless, it is widely accepted that most insulin in the brain originates in the pancreas. The rate of insulin transport across the BBB is influenced by physiological factors such as exposure to hypercaloric diets, hyperglycemia, or a diabetic state. These conditions lead to a decreased transport of insulin to the brain, and/or to a disruption of the BBB. Rhea et al., showed that IR inhibition with the selective antagonist S961, does not modify insulin transport across the BBB, suggesting that the hormone is able to cross this barrier in an IR signaling-independent manner [72].

In the brain, two different isoforms of IR can be found: a long isoform, IR-B, and a short predominant isoform [73], IR-A, with insulin having similar affinity and potency for both isoforms [74,75,76]. IR are expressed at high levels in different brain regions such as the hypothalamus, olfactory bulb, hippocampus, striatum, cerebral cortex, and cerebellum, where they are involved in different functions. The olfactory bulb is the region of the brain that contains the highest amount of IR, although its function in this area is not fully understood. The cortex and the hippocampus also have high levels of IR with very important functions in terms of synaptic plasticity and cognitive processes. The third region that contains the highest levels of this receptor is the hypothalamus with a central role in the regulation of glucose homeostasis and regulation of satiety pathways [77]. Most of these actions seem to be independent of glucose utilization [62,78].

### 4.1. Role of Insulin in Brain Glucose Metabolism and Feeding

Glucose is the main source of energy in the brain, reaching this organ through the GLUTs, which have numerous isoforms [6]. Most glucose uptake to brain and by neuronal and glial cells is insulin-independent and relies on the insulin-insensitive GLUT1 and GLUT3, among other less abundant region-specific GLUTs [79,80,81]. However, in brain regions related to cognition and metabolic control, such as the basal forebrain, hippocampus, amygdala, cortex, cerebellum, and hypothalamus, the inducible insulin-sensitive GLUT4 is reported to be co-expressed with GLUT3 [82,83,84]. Activation by insulin induces the translocation of GLUT4 to the membrane and is thought to improve glucose flux into neurons during periods of high metabolic demand, such as learning [85,86]. The effect of insulin and its relevance on glucose uptake in the brain is not consensual and for a long time was thought to be exclusively insulin independent. In fact, several studies from the 80′s that focused on the effect of insulin in brain glucose uptake showed that brain glucose metabolism was unaffected by insulin. Goodner et al. measured glucose uptake in fasting rats 30 min after 0.1U insulin administration [87] and observed that the brain did not increase the rate of glucose uptake, concluding that the brain was “insensitive to insulin” [88]. However, subsequent studies using 18-fluorodeoxyglucose positron emission tomography showed that insulin increases brain glucose uptake in humans, mostly marked in cortical areas [89]. In agreement with the role of insulin in regulating glucose uptake in the brain, GLUT4 colocalizes with IR in several brain regions [90].

Another important role of insulin in the indirect control of glucose homeostasis is related with the mesolimbic reward pathways on the dopaminergic neurons [91]. Insulin can alter food choices and preferences, as it increases dopamine release in the reward pathways after food restriction, an effect not seen in animals submitted to hypercaloric diets. Therefore, it has been shown that dopaminergic pathways are negatively regulated by insulin, as insulin causes the desire to consume high-calorie foods rich in sugar and fat, promoting a feeling of satiety and reward [91,92].

Beyond its role on controlling glucose homeostasis, insulin in the brain also impacts insulin sensitivity at the periphery. Heni and colleagues have found in lean men that central insulin action, achieved by an intra-nasal insulin spray application, improved peripheral insulin sensitivity measured by hyperglycemic clamp [93]. When they explored the mechanisms involved, they discovered that the vagus nerve and the parasympathetic system activation is involved in the peripheral insulin sensitivity regulation by CNS insulin. The hypothalamus appears to be one of the most important brain regions in this link [94]. Moreover, they have found an impairment in the central regulation of peripheral insulin signaling in obese people, showing the tight link between brain and peripheral insulin resistance in dysmetabolism conditions [94].

Apart from glucose homeostasis regulation, insulin and its receptors have implications in the regulation of energy balance where the hypothalamus also plays a major role [18]. Together with insulin, leptin, an adipokine produced by the adipose tissue, plays a role in the regulation of feeding behavior. Both hormones act on the hypothalamus affecting the expression of neuropeptides involved in satiety pathways [18,95]. In the hypothalamus, these hormones act in the arcuate nucleus (ARC) which is composed by two antagonistic types of neurons: the anorexigenic neurons or the appetite-suppressing neurons known as proopiomelanocortin-expressing neurons (POMC) neurons, and the orexigenic or appetite-stimulating neurons known as neuropeptide Y (NPY) and agouti-related peptide (AgRP) expressing neurons [18,95] (Figure 2). Both orexigenic and anorexigenic neurons express insulin and leptin receptors.

The POMC neurons project to the second order neurons in the paraventricular nucleus (PVN) but also to the lateral hypothalamus (LH), the ventromedial hypothalamus (VMN) and the dorsomedial hypothalamus (DMH) [96]. The PVN is responsible, in part, for the secretion of a wide range of regulatory neuropeptides and by the control of sympathetic nervous system activity to peripheral organs [18]. After a meal, POMC is cleaved in α-melanocyte- stimulating hormone (α-MSH) that is released by the POMC neurons to activate melanocortin 3 and 4 receptors (MC3/4R) in both hypothalamic and extra-hypothalamic neurons to suppress feeding and food intake and stimulate energy expenditure [97]. In general, insulin and leptin, by acting on POMC neurons, suppress feeding and promote energy expenditure (Figure 2). In fact, these hormones in POMC neurons also have a role in the regulation of peripheral glucose homeostasis, although the mechanisms involved are not so clear. The disruption of hypothalamic insulin and leptin pathways in POMC neurons, driven by insulin and leptin receptors deletion in these neurons, leads to a state of systemic insulin resistance and deterioration of glucose homeostasis [18]. This suggests that intact insulin and leptin hypothalamic pathways are required for a correct peripheral insulin and glucose homeostasis.

In contrast, in AgRP/NPy neurons, insulin regulates hepatic glucose production by inducing hyperpolarization and decreasing firing between AgRP neurons, which leads to reduced gluconeogenesis. Also, leptin directly acts on AgRP neurons at the hypothalamus to exert an inhibitory effect. As such, the crucial effect of leptin in the hypothalamus is to promote energy expenditure and inhibit food intake [18,97].

Apart from the regulation of feeding behavior and glucose homeostasis, insulin has been shown to have other functions in the brain as the regulation of cognitive functions, particularly memory. Notably, defects in insulin signaling in the brain may contribute to neurodegenerative disorders and damage of the cognitive system leading to dementia states.

### 4.2. Insulin and Cognitive Function

The role of insulin in the brain was first studied in the context of energy homeostasis, a process mainly regulated by the hypothalamus [68]. More recently, the role of insulin in other brain functions such as memory and cognition, neuronal development, and plasticity have been explored [6,98]. Insulin has a key role in the hippocampus, a brain region that expresses high levels of IR and that is involved in learning and memory [99].

The hippocampus is a complex structure in the brain located in the temporal lobe. It forms part of the limbic system, being an extension of the cerebral cortex [100]. The hippocampus is a very plastic region, with a key role in memory formation, organization, and storage [100]. The hippocampus converts short-term memory into long-term memory, solves spatial memory, and recollects the past experiences of places. It also plays a pivotal role in emotions and behavior [101]. Different parts of the hippocampus have distinct functions in certain types of memory: spatial memory, mainly processed by the rear part of the hippocampus; memory consolidation, a process by which the hippocampus organizes the stored information in the neocortex; and memory transfer, since long term memories are not stored in the hippocampus [101]. Information in the hippocampus travels along a unidirectional trisynaptic circuit originating from the entorhinal cortex and projecting to the dentate gyrus (DG), then to area CA3, and finally to area CA1 of the horn of Ammon [102,103]. These areas of the hippocampus have different functions, being composed of different types of neurons: the granular cells in the DG and pyramidal cells in the areas CA1 and CA3 of the horn of Ammon with a vast network of interneurons [102]. Special attention has been paid to the CA3 region for its specific role in memory formation and neurodegeneration [104]. CA3 receive excitatory inputs from the pyramidal cells and then inhibitory feedback that inhibit the pyramidal cells. This recurrent inhibition is a simple feedback circuit that can dampen excitatory responses in the hippocampus, being involved in memory formation processes [105,106].

The hippocampus has also an important role in brain plasticity. Brain plasticity underlies learning and memory and depends both on the activity and the number of synapses. Synapses may be modulated via potentiation or depression, processes that regulate the formation of new dendritic spines promoting tasks, learning, and consolidating behavioral alterations [6,107]. It has been postulated that insulin may be involved in the regulation of synaptic plasticity mechanisms and memory formation, since the mRNA levels of IR in dendrites and synapses at the hippocampus is high [99,108]. In fact, hippocampal neurons treated with insulin for 48 h show an increase in the frequency of miniature excitatory postsynaptic currents (mEPSCs), whilst downregulation of IR with short hairpin RNAs (ShRNAs) leads to the formation of few dendritic spines and therefore to reduced frequency of mEPSCs [6,109]. Insulin was also shown to regulate neuronal plasticity by controlling long-term potentiation (LTP) and long-term depression (LTD) [110]. Moreover, IR substrate p53 (IRSp53) interacts with the postsynaptic density protein 95 (PSD-95), present at excitatory synapses, therefore regulating a variety of receptors and channels, and increasing dendritic spine formation [6,111]. Insulin further plays a role on glutamatergic response by increasing the recruitment of the N-methyl-D-aspartate receptors (NMDARs) to the membrane and by enhancing the phosphorylation of NR2A and NR2B subunits [112]. In agreement, IRS2 knockout mice showed lower activation of NR2B subunits and a decrease in the LTP at CA3-CA1 synapses, however with higher density of CA1 dendritic spines [112,113,114]. Additionally, downregulation of a-amino-3-hydroxy-5-methyl-4-isoxazolepropionic acid receptors (AMPARs) activity of CA1 neurons in the hippocampus is crucial to insulin-induced LTD, an important feature to process memory formation [115].

The hippocampus is not the single player in the generation and regulation of memory and cognition. The prefrontal cortex is known to mediate decision making being involved in the retrieval of remote long-term memory and supporting memory and consolidation in a time scale ranging from seconds to days [116]. In fact, the hippocampus and prefrontal cortex work together to support the rapid encoding of new information, consolidation, and organization of memory networks. The cerebral cortex is known to process information about objects and events that we experience, and about the places where they occurred. Additionally, the ventral hippocampus (in the rat) and the anterior hippocampus (in humans) sends information to the medial prefrontal cortex (mPFC), suggesting that the mPFC could accumulate interrelated memories. The information of mPFC is sent back to other cortical regions, however the mPFC may bias or select the retrieval of event information in the ‘what’ stream. Therefore, interactions between these two brain regions can support the ability to create contextual representations that are associated with recent memories and use them to remember memories that are appropriate within a given context [117].

Considering the key role of the cortex in memory and decision making, it was reported that patients with frontal lobe damage exhibit inappropriate social behaviors and memory decline [89], suggesting that these dysfunctions can result from an impairment of memory storage in the prefrontal cortex. In fact, insulin has also an important role in the cortex, specially the IGFs since they are involved in the process of dendritic elaboration and to the stimulation of neurite outgrowth from dissociated neurons [118].

Therefore, the impairment of insulin signaling in the brain will have a negative impact in some neuronal functions additionally to the ones related with metabolism. These defects in insulin action in the CNS and specially in the hippocampus and prefrontal cortex represent a possible relationship between metabolic and cognitive disorders [119,120].

## 5. Metabolic Syndrome and the Neurodegenerative Process

It is consensual that hypercaloric diet consumption leads to obesity and several comorbidities such as insulin resistance, which can be in the genesis of T2D. The defects in insulin signaling also affects the CNS, since obesity and T2D are clear risk factors for several pathologies related with the CSN, like AD and PD [120]. It was observed that patients with T2D and/or obesity, show decreased concentrations of insulin in the cerebrospinal fluid, despite having high insulin levels in plasma [121,122]. Moreover, obesity and inflammation are reported to decrease the transport of insulin to the brain through the BBB, and T2D patients show a decreased expression of IR in the brain [121].

It was described by García-Cáceres and colleagues that alterations in insulin signaling in the brain promotes several alterations not only in neural but also in glial cell function [123]. In fact, insulin action in the brain has a direct effect on neurodegenerative and psychiatric disorders, which include alterations in dopaminergic signaling, hippocampal synaptic plasticity, expression of some proteins, and BBB function, among others [123,124].

In addition, changes in insulin action at the level of the hippocampus affect molecular mechanisms involved in synaptic formation and plasticity, having a negative impact in the maintenance of mental abilities and being a great risk factor for dementia [125]. In states of insulin resistance, IR-dependent molecular cascades may start to be insensitive to this hormone, leading to brain insulin resistance (BIR), and in this situation, insulin lose its ability to improve plasticity [122]. It is well described by different authors that rodents submitted to HF diet is a well-established animal model to study metabolic disorders since it induces metabolic features that resemble T2D in humans. Moreover, it was also described that HF diet leads to alterations in functional and structural brain plasticity, a characteristic similar to other models of insulin resistance [126]. Insulin signaling impairment promoted by a HF diet leads to the decreased expression of a post-synaptic protein, PSD-95, which has a role in promoting maturation and regulation of synaptic strength and plasticity [127]. Moreover, studies performed in Zucker rats, which are known to be insulin resistant, also showed altered hippocampal insulin signaling that was negatively associated with synaptic activity, since those animals presented impairment of LTP at CA3-CA1 synapses [128].

Several studies explored the relationship between obesity and/or T2D and neurodegenerative disorders such as PD and AD, since it was already established that people who develop T2D at an early age have an increased risk to develop neurodegenerative disorders (Figure 3) [6,122,129,130,131].

Neurodegeneration is the progressive loss of structure or function of neurons, which may ultimately involve cell death and is considered an age-related process. Neurogenerative diseases are often associated with the misfolding and aggregation of given proteins. AD is the most common neurodegenerative disease, and its pathological hallmark is the extra-neuronal accumulation of beta-amyloid (Aβ) plaques and intra-neuronal hyperphosphorylation of tau protein and formation of tau neurofibrillary tangles [132,133]. Synucleinopathies comprise a wide group of neurodegenerative disorders with a broader spectrum of clinical presentations, which includes PD, dementia with Lewy bodies, multiple system atrophy, and pure autonomic failure. These disorders are believed to result from the pathological accumulation of neuronal or glial insoluble aggregates of alpha-synuclein (aSyn) [134,135,136,137,138,139]. Although several genetic mutations and polymorphisms are described to be associated with familiar forms of neurodegenerative diseases, the etiology of these diseases remain largely elusive and molecular mechanisms poorly characterized.

In human studies, AD, cognitive impairment, and other forms of dementia have also been linked to a HF diet, obesity, DM, and metabolic syndrome [7,140,141,142,143]. Prospective studies have shown that a diet enriched in saturated fats and refined carbohydrates prime middle-aged and elderly adults to an increased risk for neurological disorders such as AD and to a faster rate of normal age-related cognitive decline [141,144,145]. Additionally, intake of this type of diet in adolescents negatively correlates with visual-spatial learning and memory performance later on and with school performance problems, especially with self-reported difficulties in mathematics [146,147]. Moreover, in children, body mass index negatively correlates with visual-spatial intelligence [148]. Even more, healthy adults exposed to a high saturated fat diet for one week had worse performance on tasks measuring attention and speed of retrieval than they had prior to the diet [149,150]. Epidemiologic reports demonstrate not only an association between obesity caused by dietary fat intake and an increased risk for AD, but also that patients with DM have a 50 to 75% increased risk of developing AD compared to age and gender matched control groups (Figure 4) [151,152,153,154]. Furthermore, T2D or impaired fasting blood glucose are reported in 80% of AD patients [155]. Accumulating data suggests that AD is closely related to insulin resistance and the dysfunction of both insulin signaling and glucose metabolism in the brain. The evidence of systemic insulin resistance and defective insulin signaling in the brain being common features of AD transformed T2D as an important risk factor for this neurological disorder. In fact, evidence suggests that insulin resistance and T2D aggravate AD pathology and cognitive deficits [156,157,158,159]. Moreover, post-mortem studies have demonstrated central insulin signaling dysregulation in hippocampal and cortical samples from patients with both mild cognitive impairment and early AD [160,161].

The hippocampus and cerebral cortex are the brain regions most vulnerable to aggregation and accumulation of Aβ and Tau, pathological hallmarks of AD. These brain regions express high levels of IRs; however, under AD conditions it has been reported that the levels of insulin mRNA are decreased [162]. In fact, impaired insulin signaling leads to decreased PI3K/AKT signaling, leading to the overactivation of GSK-3β [163]. Consequently, the over-activation of GSK-3β results in the hyperphosphorylation of tau and to a higher production of Aβ peptides, contributing to cognitive impairment (Figure 4) [163,164,165]. Additionally, insulin deficiency leads to a decrease in GLUTs levels, leading to an impairment in glucose uptake/metabolism in the brain [22]. Decreased intraneuronal glucose uptake causes a reduction of intraneuronal generation of ATP, impairing synaptic activity and cognitive function. It also leads to a decrease in the levels of uridine diphosphate (UDP)- β-N-acetylglucosamine (GlcNAc) via the hexosamine biosynthetic pathway and, consequently, decreasing tau O-GlcNAcylation, a process that inversely regulates tau phosphorylation. An increase of tau phosphorylation induces the formation of tau oligomers, which are neurotoxic, contributing to neuronal loss and degeneration [166].

The unequivocal link between AD and T2D prompted the concept of type 3 diabetes or insulin-resistant brain state to refer to AD [142]. Importantly, epidemiologic studies point long-term hyperinsulinemia as a risk factor for dementia, while insulin administration to AD patients improves memory formation, by keeping glucose levels in the brain constant [167].

The risk for PD has been shown to increase in metabolic syndrome, with T2D and obesity being important risk factors for PD [130,168,169,170,171]. Epidemiological studies revealed that 80% of PD patients have impaired glucose metabolism. Moreover, T2D can increase the risk of developing PD up to 50% in the general population and significantly accelerates the progression of both motor and cognitive deficits [9,131,172]. Impressively, this risk increases up to 380% for young diabetic individuals [130,131]. Predictably, obesity also affects motor control capabilities, degrading daily functions and health [171]. Children who are obese or overweight are poorer in gross and fine motor control and have delayed motor development [173,174,175,176,177,178,179]. PD is characterized by motor and non-motor symptoms, being a disease deeply associated with aSyn aggregation, accumulation and toxicity, and dopaminergic neuronal loss in the substantia nigra pars compacta [180].

Following the strong evidence of a link between metabolic syndrome and neurodegenerative diseases dissected in clinical and epidemiological studies, animal models of human diseases have been largely used to scrutinize the molecular mechanisms and pathways underling this association. An extensive description of these studies will be provided below.

### 5.1. The Impact of Diet in Neurodegeneration: Evidence from Animal Models

Evidence provided by exhaustive research using animal models strongly suggests that diet plays a crucial role in brain physiology and pathology. Currently, the research field is based on both pharmacological and genetic model of AD and PD, both in rats and mice, that have been crucial to disclose molecular mechanisms and dysregulated pathways associated to these neurodegenerative disorders. In the case of metabolic syndrome, as previously described, researchers can also count with diet-induced and genetic models of the disease.

#### 5.1.1. Parkinson’s Disease Animal Models

The most accepted and used pharmacological models of PD use 6-hydroxydopamine (6-OHDA) or 1-methyl-4-phenyl-1,2,3,6-tetrahydropyridine (MPTP) to induce dopaminergic degeneration and to induce parkinsonism. Most of the work that aimed to study the impact of diet on neurodegenerative diseases has been performed by submitting these PD models to hypercaloric diets. As it can be seen in Table 1, several studies pointed towards an exacerbation of neurotoxic/neurodegenerative mechanism in the presence of hypercaloric diets, and in particular HF diet.

In pharmacological PD-like models, several studies reported that the HF diet potentiated the decrease of tyrosine hydroxylase (TH) and dopamine depletion in the substantia nigra and/or striatum, suggesting that both MPTP and 6-OHDA exacerbate dopaminergic neurodegeneration. Furthermore, a few studies correlated these alterations with motor abnormalities [181,182,183,184,185]. In transgenic models of PD, the same trend for the impact of HF diet in neurodegeneration features was observed (Table 1). In the transgenic mouse models, HF diet accelerated the onset of aSyn pathology, locomotor impairment, and enhanced dopaminergic degeneration [186,187].

In recent years, more physiological studies, based on challenging the animal models to hypercaloric diets, have been the gold standard in the field. Although beyond insulin resistance, DM, and obesity phenotypes, several studies have been reporting the effects of non-standard diets on brain function and physiology, on motor performance and cognition, and on neurodegenerative-associated processes. In more physiological models using wild type (WT) rats and mice, HF diets reduced the levels of TH in the substantia nigra and striatum and attenuated dopamine release and clearance, accompanied by a decrease in movement together with abnormal motor behavior and other behavior alterations [180,188,189,190]. In a study where no alterations were observed in the levels of aSyn, the authors claimed that the reduction of TH levels in the nigrostriatal axis occurs through an aSyn-independent pathway and can be attributed to brain inflammation, oxidative stress and metabolic syndrome induced by obesity [180].

However, even if these few studies point towards a link between hypercaloric diets-associated defects in insulin signaling and brain glucose metabolism and the neuropathological features related with PD, the mechanisms by which the hypercaloric diets impact in these neuropathological features deserves further investigation. Also, we would like to highlight, that contrary to what happens for other neurodegenerative diseases as dementia and AD, there are not so many studies evaluating the impact of hypercaloric diets per se and dysmetabolism on the neuronal function and degeneration leading to motor alteration conducting to PD, as well as the impact of different diet composition on these neuropathological features.

**Table 1 nutrients-14-01425-t001:** Effects of hypercaloric diets on Parkinson’s disease-like and wild-type rodent models.

Study	Diet Regiment	Rodent Model	Outcomes
Choi et al. [181]	8 weeks of HF diet	MPTP-lesioned PD-like mice	Severe decrease in the levels of striatal dopamine and of nigral microtubule-associated protein 2, manganese superoxide dismutase, TH. Elevated striatal nNOS phosphorylation and dopamine turnover.
Bousquet et al. [182]	8 weeks of HF diet	MPTP-lesioned PD-like mice	Decreased levels of striatal TH and dopamine, exacerbated MPTP-induced dopaminergic degeneration.
Sharma and Taliyan [183]	8 weeks of HF diet	6-OHDA-induced PD-like rats	Decreased levels of striatal dopamine, motor abnormalities, exacerbated 6-OHDA mediated neurotoxicity.
Morris et al. [184]	5 weeks of HF diet	6-OHDA-induced PD-like rats	Peripheral dysmetabolic features, increased dopamine depletion and oxidative stress in the substantia nigra and the striatum, without locomotor dysfunction.
Ma et al. [185]	3 month of HF diet, followed by 3 months of a low-fat diet	6-OHDA-induced PD-like rats	Reversed peripheral dysmetabolism and mitochondrial and proteasomal function in the striatum, although without altering nigrostriatal vulnerability.
Rotermund et al. [191]	HF diet from 5 weeks old onward throughout their lifespan	Mutant A30P aSyn transgenic mice	Accelerated onset of brainstem aSyn pathology and lethal locomotor features.
Hong et al. [192]	2 weeks of HF diet	MitoPark transgenic mice	Increased *SNCA* expression (coding for aSyn) in the dopaminergic neurons of both the WT and MitoPark mice; enhanced dopaminergic degeneration in the MitoPark mice.
Morris et al. 2011 [188]	12 weeks of HF diet	WT rats	Attenuated dopamine release and clearance and increased iron deposition in the substantia nigra.
Jang et al. [189]	13 weeks of HF diet	WT mice	Decreased in movement accompanied by abnormal motor behavior. Decreased levels of TH in the substantia nigra and striatum.
Kao et al. [193]	5 months of HF diet	WT mice	Dopaminergic neurons degeneration and reduced dopaminergic neuroplasticity in the substantia nigra.
Bittencourt et al. [180]	25 weeks of HF diet	WT rats	Reduced levels of TH through metabolic dysfunction, neuroinflammation and oxidative stress, associated with impaired locomotor activity, and anxiety-related behaviors, without changes in motor coordination or memory. No differences in the levels of aSyn.

6-OHDA, 6-hydroxydopamine; aSyn, alpha-synuclein; HF, high-fat; MPTP, 1-methyl-4-phenyl-1,2,3,6-tetrahydropyridine; PD, Parkinson’s disease; TH, tyrosine hydroxylase; WT, ild type.

#### 5.1.2. Alzheimer’s Disease Animal Models

As stated before, there is an unequivocal link between AD and T2D/impaired insulin signaling, which prompted the concept of type 3 diabetes to refer to AD. This link comes from the extensive literature reporting the deregulation of insulin signaling in AD animal models and patients. Valuable insights on the role of hypercaloric diets in cognitive function were provided by several studies, associating their intake with pathological markers of AD (see Table 2).

Velazquez and collaborators reported central insulin dysregulation and energy dyshomeostasis in two transgenic mouse models of AD, the Tg2576 and the triple transgenic AD (3xTg-AD) mice [194]. Moreover, HF diet in transgenic AD-like mice enhanced memory impairment and cognitive decline without affecting the levels of Aβ and phospho-tau protein. The exacerbated phenotype seems to be mediated by increased inflammation and oxidative stress [195,196]. Moreover, both HF or HFHSu diets in WT mice and rats are reported to induce significant cognitive alterations, mainly short- and long-term, contextual, and auditory-cued fear memory impairment. These behavioral phenotypes were associated with insulin resistance, increased neuroinflammation, Aβ deposition and neurofibrillary tangle formation, and decreased synaptic plasticity [197,198,199,200,201,202,203,204]. In agreement, Abbott and co-workers conducted a meta-analysis of the results from rodent studies using different diets (HF, HSu, or HFHSu diets) and reported that each type of diet and task adversely affected performance, with the largest effect produced by exposure to a combined HFHSu diet, as assessed by the radial arm maze to evaluate the effect of such diets on cognition [205].

Altogether, hypercaloric diets prompt to neurogenerative processes that are in the basis of cognitive dysfunction and that different diet composition conduce to different pathological mechanisms originating different neuropathological features. Moreover, several mechanisms contribute to these neurogenerative processes, namely insulin signaling dysregulation, altered brain glucose homeostasis, neuroinflammation, and oxidative stress, among others. Therefore, strategies aiming to prevent and regulate the hypercaloric diets—associated pathological mechanisms, namely the control of dysmetabolic states might be extremely useful for the prevention and delay of neurodegenerative processes.

### 5.2. Sex Differences in the Link Dysmetabolism-Neurodegeneration

Gender or sex is a significant variable in the prevalence and incidence of neurodegenerative disorders, such as AD and PD. In fact, sex dimorphisms have a cardinal role in the pathogenesis, progression, age of onset, and treatment response in AD and PD. The prevalence of AD has been reported to be higher in women than in men (1.6–3:1 female/male ratio) [206,207,208,209,210,211,212,213]. Several studies showed that the extent of cognitive deficits is higher in AD females, with enhanced clinical expression of disease pathology due to more neurofibrillary tangles [214]. In contrast, the prevalence of PD is reported to be higher in men than women (1:1.5–3 female/male ratio) [215,216,217,218]. Studies reveled that PD symptoms like rigidity, rapid eye movement behavior disorder, sleep disturbances, deficits in verbal communication, and lack of facial emotions are more prevalent in males. On the other hand, the onset of PD is delayed in females, that exhibit a tremor-dominant form of the disease with greater impairment of postural stability, depression, and reduced ability to conduct daily activities, and in which reduction in visuospatial cognition is more frequent [214].

Although the biological factors and molecular mechanisms underlying these gender disparities are still poorly understood, sex steroid hormones, including estrogens, progestogens, and androgens, have been implicated in these sexual dimorphisms. Notably, sexual steroid hormones are known to regulate energy metabolism [219,220,221,222,223] and exert neuroprotective effects on the adult brain, increasing neural function and resilience and promoting neuronal survival [224,225,226,227]. The aging process is characterized by a shift in the hormonal profile both in men and women, with an abrupt loss of estrogens during menopause in women and a gradual but significant decline of testosterone during andropause in men [219,224]. Despite the brain levels of steroid hormones depend on both endocrine gland production and on the local synthesis of neurosteroids, the brain also accounts with a decline in the levels of sex steroids [228]. Thus, reproductive senescence has been implicated in metabolic alterations and negatively impacts neural function and represents a significant age-associated risk factor for AD and PD.

## 6. Regulation of Metabolic Function as a Prevention of Neurodegeneration

The existing therapies for neurodegenerative disorders are not disease-modifying. For the case of PD, they are only able to alleviate the symptoms. They are mostly dopamine-replacing therapies (levodopa-carbidopa, dopamine agonists), able to improve PD-motor features in the initial stages of the disease. Tackling non-motor features require non-dopaminergic approaches as for example, the selective serotonin reuptake inhibitors [229]. For the case of AD, current treatments also aim to attenuate symptoms (e.g., NMDA receptor antagonists and drugs that target cholinergic transmission) such as cognitive impairment, aggression, and seizures, and do not target disease pathology [230].

Considering the increased risk for neurodegenerative diseases among patients with T2D and the remarkable pathophysiological mechanisms shared, there is mounting interest in the potential of antidiabetic agents for the treatment of these neurological disorders. Hundreds of pre-clinical and clinical studies have explored the potential of antidiabetic medications both for lowering the risk of and as a novel therapeutic avenue for neurodegenerative diseases such as PD and AD.

In recent years, several anti-diabetic drugs have been investigated in PD and AD patients, including metformin, glucagon like peptide (GLP-1) analogues, dipeptidyl peptidase 4 (DPP-4) inhibitors or gliptins, among others. For example, a study performed in a Taiwanese cohort evaluated the usage of metformin, known to reduce blood glucose levels and to decrease insulin resistance, in PD patients and reported that the combination of metformin with sulfonylurea therapy was able to reduce the risk for PD [231,232]. Indeed, findings from human studies are recapitulated in animal models. Metformin administered to MPTP-induced PD mice was shown to improve locomotor activity in these animals [233]. Metformin was also demonstrated to be able to cross the BBB and to act as a neuroprotective agent by reducing tau phosphorylation in primary cortical neurons of a transgenic mouse with the lack of microtubule-associated protein tau (MAPT), a known mechanism of AD development [234]. Moreover, metformin is also reported to have anti-inflammatory properties, to lower reactive oxygen species (ROS) production, and to act as methylglyoxal scavenger and an advanced glycation end products (AGEs) inhibitor [235,236,237,238,239]. Metformin was shown to prevent amyloid plaque deposition and memory impairment in APP/PS1 mice, to restore neuronal insulin signaling and prevent AD-associated pathological changes in neuronal cultures and to protect against Aβ-induced mitochondrial dysfunction [240,241,242]. Metformin prevents aSyn accumulation, aggregation, and phosphorylation, and prevents or attenuates dopaminergic neurodegeneration, ameliorates pathology, and improves locomotor activity and coordination in acute MPTP- and 6-hydroxydopamine (6-OHDA)-induced Parkinsonism rodent models [233,243,244,245,246,247]. Additionally, a Singapore longitudinal aging study showed that long-term treatment with metformin is associated with reduced risks of cognitive decline in T2D patients. [248]. Controversially, studies conducted by Imfeld and Moore reported a slightly higher risk of AD and cognitive impairment in patients with T2D under metformin therapy [249,250]. Furthermore, Kuan, Ping, and Qin showed overall lack of correlation between metformin therapy and the prevention of PD and reported that long-term metformin exposure in patients with T2D may increase the risk for PD and dementia [251,252,253]. These conflicting results may derive from the high heterogeneity among clinical studies in terms of population, treatment plan, and follow-up. Furthermore, the effect of metformin likely depends on complex neurodegeneration-associated pathological processes and may not be beneficial for all patients with PD or AD, rather than for patients with evidence of metabolic disease.

Thiazolidinediones (TZDs) are insulin sensitizers that primarily reduce insulin resistance in insulin-sensitive tissues in the periphery. Promising results from preclinical studies reported that pioglitazone and rosiglitazone are able to ameliorate memory and cognitive impairment and decrease AD-related pathology in several rodent models of AD, including the 3xTg-AD mice and Tg2576 mice, by enhancing AKT signaling and attenuating tau hyperphosphorylation, neuroinflammation, and AD pathology [254,255,256,257]. Furthermore, pioglitazone, rosiglitazone and mitoglitazone were also shown to prevent dopaminergic neurodegeneration and locomotor deficits in MPTP-, STZ-, and 6-OHDA-lesioned rodent models of PD, by modulating the neuroinflammatory response in a neuroprotective fashion [258,259,260,261,262,263,264]. These findings are consistent with data from clinical studies. TZDs lead to an improvement in memory and selective attention in patients with AD and amnestic mild cognitive impairment [265,266,267].

Another class of antidiabetic drugs used to manage the neurodegenerative process are the sulfonylureas, which act to increase insulin release from the beta cells in the pancreas. In fact, several studies support the idea that sulfonylureas could be used as therapeutics to AD and PD. For example, the use of glimepiride was shown to protect against Aβ induced synaptic damage in neuronal cultures [268]. Also, T2D patients treated with sulfonylureas exhibit low risk of dementia, and a combination of metformin with sulfonylureas decreased the risk by 35% over 8 years [269].

Sodium-glucose cotransporter 2 (SGLT2) inhibitors, the newest class of anti-diabetic drugs used in T2D treatment, act via the inhibition of renal glucose reabsorption therefore promoting glucose excursion [270]. SGLT2 inhibitors have neuroprotective actions in T2D mouse models, particularly by reducing inflammation and oxidative stress [271,272,273]. In agreement, empagliflozin reduced cerebral oxidative stress and impairment of cognitive function in db/db animals [271]. Moreover, dapagliflozin improved cognitive decline in HF diet animals [273].

Another class of anti-diabetics are the modulators of incretins, that include GLP-1 receptor agonists and DPP-4 inhibitors (vildagliptin or sitagliptin) used for T2D and obesity (in the case of GLP-1 agonists). Dulaglutide, a GLP-1 receptor agonist, ameliorated STZ-induced AD-like impairment of learning and memory ability by modulating hyperphosphorylation of tau and neurofilaments [274]. Liraglutide, also a GLP-1 receptor agonist, prevented the accumulation of Aβ plaques, decreased tau hyperphosphorylation and neurofilament proteins, prevented the loss of brain IR and synapses, and reversed cognitive and memory impairment in several mouse models of AD-like pathology and in a non-human primate model of AD [275,276,277]. In relation to PD, a huge amount of evidence supported the repurpose of the use of GLP-1 agonists for the treatment of this pathology [278,279]. Exenatide and Ex4 both in animal models and humans was shown to promote neuroprotection (for a review see [278,279]). For example, these drugs in human trials led to an improvement of motor symptoms, to benefits in cognitive performance, as well as to better results in tremor-dominant phenotype [280,281,282,283]. Liraglutide and lixisenatide ameliorated locomotor function and prevented dopaminergic neurons degeneration, by suppressing neuroinflammation in the substantia nigra in the MPTP-induced PD-like mice and rotenone-lesioned PD-like rats [284,285,286]. Also, semaglutide in chronic MPTP mouse model of PD showed to promote neuroprotection by decreasing aSyn levels and neuroinflammation with consequent improvement in motor function [287,288].

Sitagliptin, saxagliptin and vildagliptin, DPP-4 inhibitors, improves learning and memory deficits through decreasing abnormal phosphorylation of tau and neurofilaments (NFs), reducing intercellular Aβ accumulation in 3xTg-AD and STZ-lesioned mice [289,290,291]. Sitagliptin reversed nigrostriatal degeneration, improved motor performance, and rescued the memory deficits in rotenone-lesioned PD-like rats [286,292]. The use of DPP4 inhibitors and GLP-1 mimetics is associated with a lower risk for PD and AD among patients with T2D, even compared to the use of other oral antidiabetic drugs [293,294]. All together, these clearly state that the decrease in insulin resistance and the regulation of whole-body glucose homeostasis, and therefore a better metabolic control, is associated with better cognitive function and slower deterioration of brain capacities related to the neurodegenerative process.

Another way to improve metabolic control is by targeting the carotid bodies (CBs), metabolic sensors deeply involved in peripheral insulin action and glucose homeostasis [295,296,297]. The CBs are peripheral chemoreceptors located near the bifurcation of the common carotid artery, classically defined as oxygen, carbon dioxide, and pH sensors [298,299]. They are surrounded by a dense net of capillaries and penetrated by the sensory nerve ending of the carotid sinus nerve (CSN) a tiny branch of the glossopharyngeal nerve which does the connection from the CBs to the brainstem where their information is integrated [298,300]. In recent years, the CBs have been shown to have a key role in the control of peripheral glucose metabolism and insulin action, since it was shown that: (1) in animal models of dysmetabolism and in prediabetic patients, the CBs were overactivated, this being correlated with a state of insulin resistance and glucose intolerance [295,301,302]; (2) insulin, leptin, and GLP-1, known metabolic mediators involved in metabolic regulation, are capable of activating the CBs [295,296,303,304]; and (3) the resection or neuromodulation of the CSN prevents and reverses the pathological features associated with dysmetabolic states [295,297,301]. Considering the close association between metabolic and neurodegenerative disorders, and that the therapeutics that provide metabolic control have a huge impact on neurodegenerative processes, we anticipate that the modulation of the CBs might be a useful therapeutic strategy to improve metabolism, and therefore prevent and/or delay the progression of neurodegenerative disorders.

## 7. Conclusions

The association between dysmetabolic pathologies, like T2D or obesity, and neurodegenerative disorders is unequivocal. We have compiled the most important literature discussing the pathological mechanisms behind this relationship. Altogether it shows that the increasing incidence of both types of diseases is somehow related with worldwide aging and life habits changing, with special attention to overnutrition and low quality of nutrient intake. Insulin also has a clear central role in the relationship between metabolic and neurodegenerative disorders since it joins important peripheral and central actions for glucose homeostasis, cognitive function regulation, and food behavior control. However, it is still unclear which disease is cause or consequence. We reason that it is instead a vicious cycle. The covered literature also opens doors for possible new targets of therapeutic interventions, both by pharmacological (antidiabetic drugs) or non-pharmacologic (CBs modulation) approaches, to control these epidemics of metabolic and neurodegenerative diseases.

## Figures and Tables

**Figure 1 nutrients-14-01425-f001:**
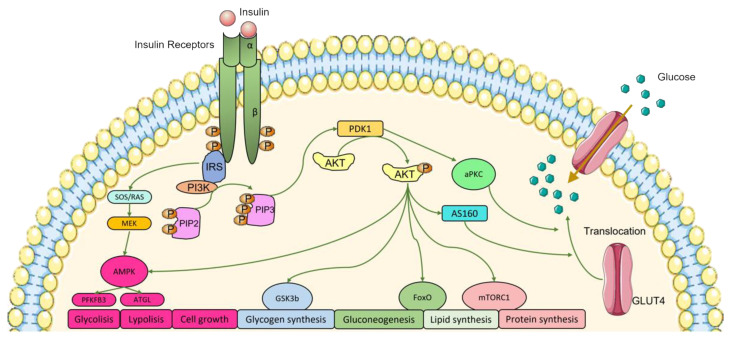
Schematic representation of insulin signaling pathways. When insulin binds to its receptor (IR), autophosphorylation of tyrosine kinase residues occurs and various downstream regulatory proteins are recruited. The insulin signal is transduced among the target proteins/enzymes ending with the fusion of the glucose transporter 4 (GLUT4) vesicle with the cell plasma membrane and the placement of GLUT4 transporters in the plasma membrane leading to the uptake of glucose. IRS, insulin receptor substrate; PI3K, phosphatidylinositide-3-kinase; PIP2, phosphatidylinositol (4,5)-biphosphate; PIP3, phosphatidylinositol (3,4,5)-triphosphate; PDK1, phosphoinositide-dependent protein kinase-1; AKT, protein kinase B; AS160, Akt substrate of 160 kDa; aPKC, atypical protein kinase C; SOS/RAS, son of sevenless; MEK, mitogen-activated protein kinase; AMPK, mitogen-activated protein kinase; PFKFB3, 6-phosphofructo-2-kinase/fructose-2,6-biphosphatase 3; ATGL, adipose triglyceride lipase; GSK3b, glycogen synthase kinase b; FoxO, forkhead box protein O1; mTORC1, mammalian target of rapamycin complex 1; GLUT4, glucose transporter type 4.

**Figure 2 nutrients-14-01425-f002:**
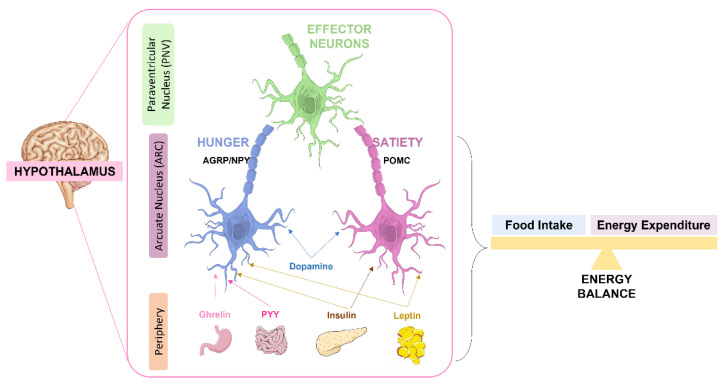
Role of the hypothalamus in the regulation of energy balance. Signals as leptin and insulin act antagonistically in the two antagonistic neurons of the arcuate nucleus (ARC): the orexigenic (appetite-stimulating) neuropeptide Y (NPY) and agouti-related peptide (AgRP)-expressing AgRP/NPY neurons and the anorexigenic (appetite-suppressing) proopiomelanocortin (POMC)-expressing POMC neurons. By one way, insulin and leptin stimulate POMC neurons, in another way they inhibit AgRP/NPY neurons. Both AgRP/NPY neurons and POMC neurons project to second-order neurons in the paraventricular nucleus (PVN), leading to an integrated response on energy intake and expenditure. The intestine also secretes peptide YY (PYY) and the stomach ghrelin that act on AgRP/NPY neurons to stimulate hunger. Dopamine modulates both AgRP/NPY and POMC neurons.

**Figure 3 nutrients-14-01425-f003:**
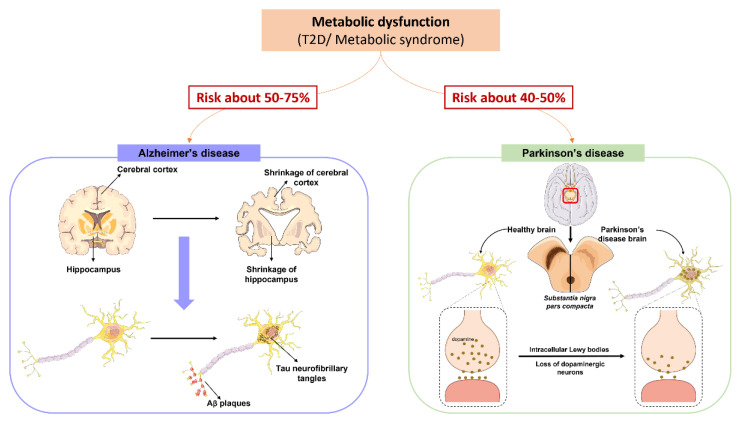
Metabolic syndrome increases the risk of developing Alzheimer’s and Parkinson’s diseases. **Left** and **right** panels show, respectively, the pathophysiology of Alzheimer’s (AD) and Parkinson’s (PD) diseases. The **left** panel, in violet, shows the physiological structure of a healthy brain and an Alzheimer’s disease brain, presenting extracellular accumulation of Aβ plaques and intraneuronal accumulation of neurofibrillary tangles of hyperphosphorylated tau protein. The **right** panel, in green, shows the physiological structure of a healthy substantia nigra versus a substantia nigra of PD patient. In PD there is the aggregation and accumulation of aSyn in Lewy bodies which are toxic leading to dopaminergic neuronal loss.

**Figure 4 nutrients-14-01425-f004:**
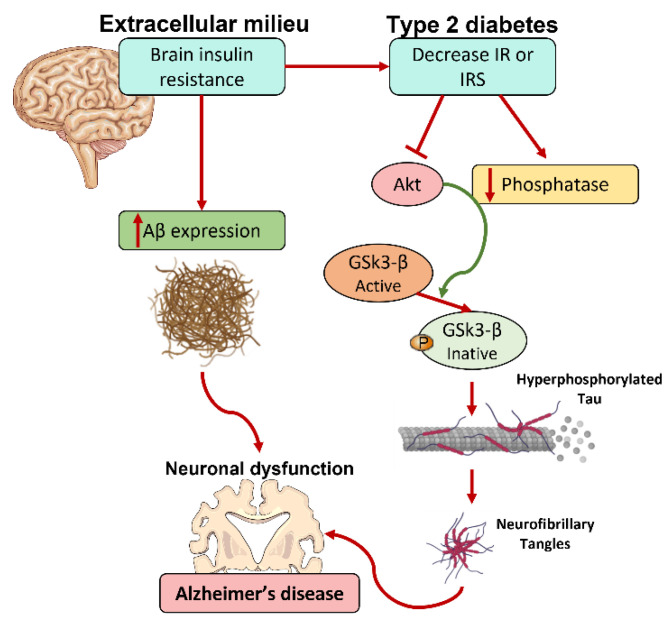
Type 2 diabetes accelerates Alzheimer’s disease pathology. Brain insulin resistance leads to alterations in the insulin signaling pathway that inactivate AKT triggering the inactivation of GSK3-β leading to tau hyperphosphorylation and higher production of Aβ peptides. AKT, Protein kinase B; GSK-3β, Glycogen synthase kinase 3 beta; IR, insulin receptor; IRS, insulin receptor substrate.

**Table 2 nutrients-14-01425-t002:** Effects of hypercaloric diets on Alzheimer’s disease-like and wild-type rodent models.

Study	Diet Regiment	Rodent Model	Outcomes
Velazquez et al. [194]	NC diet	Tg2576 and 3xTg-AD transgenic AD-like mice	Central insulin dysregulation and energy dyshomeostasis, developed before peripheral insulin resistance.
Sah et al. [195]	16 weeks of HF diet	3xTg-AD transgenic AD-like mice	Enhanced memory impairment, without alteration in the levels of Aβ and phosphorylation of tau in the cortical region. Increased neuronal oxidative stress and apoptosis.
Thériault et al. [196]	4 months of a HF diet	APPswe/PS1 transgenic AD-like mice	Accelerated age-associated cognitive decline without affecting parenchymal Aβ. Loss of synaptic plasticity and exacerbated systemic inflammation and oxidative stress.
Valladolid-Acebes et al. [197]	8 weeks of HF diet	WT mice	Spatial memory impairment and changes in hippocampal morphology, accompanied by an increase of dendritic spine density in CA1 pyramidal neurons that correlated with the upregulation of neural cell adhesion molecule (NCAM) in this area and a desensitization of the Akt pathway coupled to hippocampal leptin receptors.
Ledreux et al. [198]	6 months of HF or high cholesterol diet	WT rats	Memory impairment, neurodegeneration in the hippocampus, increased activation of microglia and abnormal phosphorylation of Tau.
Busquets et al. [199]	15 months of a HF diet	WT mice	Long-term exposure to HF diet favors the appearance of Aβ depositions in the brain, thought increased inflammation leading to a decrease in the neuronal precursor cells, and dysregulation in normal autophagy and apoptosis.
Tran and Westbrook [200]	HFHSu and NC diets	WT rats	Impairment in place-recognition memory, that is reversible and training-dependent.
Spencer et al. [201]	3 days of HF diet	WT rats	Impaired long-term contextual (hippocampal-dependent) and auditory-cued fear (amygdalar-dependent) memory in aged, but not young adult rats. Increased activation of microglia.
Kothari et al. [202]	14 weeks of HFHSu diet	WT mice	Induced brain insulin resistance, accompanied by inflammatory and stress responses as well as by increased Aβ deposition and neurofibrillary tangle formation, and decreased synaptic plasticity and cognitive impairment.
Fu et al. [203]	6 months of HF diet	WT rats	Induced hippocampal microvascular insulin resistance and cognitive dysfunction.
Fazzari et al. [204]	12 weeks of HF diet	WT hamsters	Reduced locomotor activities such as exploratory bouts, rearing and grooming behaviors, cognitive and memory impairment.

AD, Alzheimer’s disease; HF, high-fat; HFHSu, High-fat–high-sucrose; NC, normal chow; WT, wild-type.

## Data Availability

Not applicable.

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
