# Peer review of "Dysmetabolism and Neurodegeneration: Trick or Treat?"

_nutrients, 2022, doi:10.3390/nu14071425_

Round 1
Reviewer 1 Report
Cr Nutrients-1632476
Dysmetabolism and neurodegeneration: trick or treat?
The review is concise, rigorous, well referenced and addresses a topic of unquestionable relevance.
However, several aspects could be addressed:
- To introduce data in reference to sex, dysmetabolism and neurodegeneration. In this sense, the authors could include data from humans, and also about experimental studies (lines 177-181, pages 4-5).
- The authors could include several comments in reference to insulin resistance / renin–angiotensin system, or renin–angiotensin–aldosterone system, recently involved in neurodegenerative processes. Also, they could include their idea about the relationship between renin–angiotensin system and diabetes mellitus (T2D).
- In a similar line, aspects considered in these two references could be very interesting to take in account:
Front Neuroendocrinol. 2021 Jul;62:100914. doi: 10.1016/j.yfrne.2021.100914.
Neural Regen Res . 2022 Aug;17(8):1652-1658. doi: 10.4103/1673-5374.332122.
- Line 594 (page 14). Something is wrong.
Author Response
The authors acknowledge the reviewer comments and have answered to all points raised by the reviewer. We hope that in the present form the manuscript is suitable for publication. Please see responses below.
To introduce data in reference to sex, dysmetabolism and neurodegeneration. In this sense, the authors could include data from humans, and also about experimental studies (lines 177-181, pages 4-5).
A new chapter in section 5 related with the sex differences in the link dysmetabolism- neurodegeneration was now included.
The authors could include several comments in reference to insulin resistance / renin–angiotensin system, or renin–angiotensin–aldosterone system, recently involved in neurodegenerative processes. Also, they could include their idea about the relationship between renin–angiotensin system and diabetes mellitus (T2D).
The authors acknowledge the reviewer suggestion. In fact, we know that besides endocrine forms there are local forms of RAS in the brain, playing essential roles in brain homeostasis. RAS-related receptors dysfunction also has been associated with aging dysregulation leading to adverse clinical outcomes such as Alzheimer’s disease (AD) and other neurological disorders via excessive oxidative stress, neuroinflammation, endothelial dysfunction, microglial polarization, and alterations in neurotransmitter secretion (doi: 10.1093/brain/aww341; doi: 10.1212/WNL.0000000000006949; doi.org/10.3389/fnins.2020.586314). At the same time, it has been suggested that RAS may be involved in the link AD and insulin resistance (doi: 10.1055/s-0029-1212088) by many different mechanisms.
However, since the scope of this manuscript is to discuss the impact of diets and therefore dysmetabolism in the brain, in the context of the link neurodegenerative disorders-insulin resistance in a broad way, and that the manuscript is already a little bit extensive, we think that the involvement of RAS in this context it would fit in a manuscript focused more on a mechanistic perspective. Therefore, we decided not to include the suggestion of the reviewer, but we will keep it in mind for a future mechanistic approach of discussing this thematic.
In a similar line, aspects considered in these two references could be very interesting to take in account:
Front Neuroendocrinol. 2021 Jul;62:100914. doi: 10.1016/j.yfrne.2021.100914.
Neural Regen Res . 2022 Aug;17(8):1652-1658. doi: 10.4103/1673-5374.332122.
Both references and additional information regarding the studies on the impact of GLP1 agonists in PD animal models and humans have been included in the present version.
Line 594 (page 14). Something is wrong.
The reviewer is right and now it can be read “In a study where no alterations were observed in the levels of aSyn, the authors claimed that the reduction of TH levels in the nigrostriatal axis occurs through an aSyn-independent pathway and can be attributed to brain inflammation, oxidative stress and metabolic syndrome induced by obesity [180].”
Reviewer 2 Report
Dear Editor,
The manuscript entitled “Dysmetabolism and neurodegeneration: trick or treat?” by Capucho et al. describes the potential link between metabolic syndrome and neurodegeneration. It starts with an overview of insulin signalling pathways and associated pathologies outside of the brain. The authors then draw the link to insulin function in the brain and neurodegenerative diseases such as Alzheimers and Parkinson’s Disease. The manuscript is well written, easy to read and of interest for the readers of Nutrients.
I have a few minor points though:
- The authors should define clearly in the introduction what they mean when they talk about metabolic dysfunction. Anything can be a metabolic dysfunction, e.g. an acid-base imbalance, dysfunctional iron, calcium, glucose, or lipid metabolism or an inborn error of metabolism. I assume the authors mean metabolic syndrome when they write metabolic dysfunction. It is a similar problem when people talk about mitochondrial dysfunction, because it is always assumed to be related to oxidative phosphorylation, but the mitochondria has many more functional roles.
- I never heard the phrase “opened a black door” in line 49, I am not a native speaker though.
- Abbreviations used in the figures should be listed in the figure captions
Author Response
The authors acknowledge the reviewer comments and have answered to all minor points raised by the reviewer. We hope that in the present form the manuscript is suitable for publication. Please see responses below.
- The authors should define clearly in the introduction what they mean when they talk about metabolic dysfunction. Anything can be a metabolic dysfunction, e.g. an acid-base imbalance, dysfunctional iron, calcium, glucose, or lipid metabolism or an inborn error of metabolism. I assume the authors mean metabolic syndrome when they write metabolic dysfunction. It is a similar problem when people talk about mitochondrial dysfunction, because it is always assumed to be related to oxidative phosphorylation, but the mitochondria has many more functional roles.
We understand the comment of the reviewer and we have now clarified throughout the manuscript.
- I never heard the phrase “opened a black door” in line 49, I am not a native speaker though.
We removed the word “black” to be clearer for everyone and hope that in the present form the sentence is clearer.
- Abbreviations used in the figures should be listed in the figure captions
We have included all the abbreviations in figure captions.
Reviewer 3 Report
Check my comments throughout the text

Author Response
We thank the reviewer for her/his comments. We have adequately addressed the comments made by the reviewer.
Page 13 This title should fit in the next page. Done
Page 14 Some extra space should be added here. Done
Page 14 This table should all fit in the same page. Done
Page 15 Table 2 should fit in the next page and not be split between 2 different pages. Done